# Research on Modulation Signal Recognition Based on CLDNN Network

**Binghang Zou** , **Xiaodong Zeng * and Faquan Wang**

School of Electrical Engineering, Sichuan University, Chengdu 610017, China; zoubinghang@stu.scu.edu.cn (B.Z.); wangfaquan@stu.scu.edu.cn (F.W.)
* Correspondence: zengxiaodong@scu.edu.cn

**Abstract:** Modulated signal recognition and classification occupies an important position in electronic information warfare, intelligent wireless communication, and fast modulation and demodulation. To address the shortcomings of existing recognition methods, such as high manual involvement, few recognition types, and a low recognition rate under a low signal-to-noise ratio, we propose an attention mechanism short-link convolution long short-term memory deep neural networks (ASCLDNN) recognition model. The network is optimized for modulated signal recognition and incorporates an attention mechanism to achieve higher accuracy by adding weights to important signals. The experimental results show that ASCLDNN can recognize 11 signal modulations with high accuracy at a low signal-to-noise ratio and no confusion for specific signals.

**Keywords:** automatic modulation recognition; deep learning; CLDNN; attention mechanism

## 1. Introduction

The main purpose of modulation signal recognition is to extract the key features of the obtained signal, such as high-order cumulant [1] and cyclic spectrum feature [2], and then use the classifier for classification and recognition to provide a basis for further analysis and processing. At present, various recognition methods rely on manual participation for feature extraction and cannot deal with the signal types in complex situations. Therefore, it is necessary to find a new method, which can not only automatically extract the deep features of the signal to improve the accuracy, but also has high recognition accuracy when the signal-to-noise ratio is low.

Deep learning has the outstanding advantage of being able to automatically perform feature extraction on signals and has been widely used in the field of modulated signal identification in recent years. Ref. [3] carried out a study on machine learning based on convolutional neural networks for radio signal modulation. Ref. [4], on the other hand, investigated the deep structure of modulation recognition and obtained an improvement in the effect. Ref. [5] discussed the advantages of radio signal classification based on aerial deep learning compared to traditional algorithms. Ref. [6] trained different neural networks, such as baseline neural network and residual neural network, and obtained relatively high recognition rates. Ref. [7] optimized the convolutional network model with three convolutional layers and one LSTM layer, and preprocessed the higher order accumulation of each signal and obtained good results. Ref. [8] directly used LSTM networks without preprocessing the signals, and the overall results were good, but the results were poor at low signal-to-noise ratios because the LSTM layers would lose some timing information. Ref. [9] used a one-dimensional convolutional long short-term deep neural network (1CLDNN) model combined with a centerloss loss function to classify the radiation source signal, and the signal recognition accuracy was improved compared with the convolutional neural network (CNN) model, but the recognition accuracy was limited in the low SNR environment; ref. [10] proposed to improve the CLDNN model by combining the Bi-GRU [11] by replacing the long short-term memory (LSTM) layer in the

CLDNN network with a bidirectional structure, which improves the recognition accuracy and recognition speed, but still does not work well for more complex low SNR signals.

In this paper, to address the problems of a low recognition rate and signal confusion in the existing recognition method CLDNN model for radio-modulated signal recognition, the ASCLDNN (attention-mechanism short-link convolution long short-term memory deep neural networks) network is used. The short-link layer, which is more effective in recognizing modulated signals, is used in the CLDNN network. The attention mechanism is added on top of it to assign higher weights to important information and improve the network performance. We test the recognition effect of this model under various cases and find the network parameters with better effect.

## 2. CLDNN Model

CLDNN (convolutional, long short-term memory, fully connected deep neural networks) networks usually connect several layers of CNN after the input layer to reduce the frequency domain change; the output of the CNN layer enters several layers of LSTM to reduce the time domain change, and the output of the last layer of LSTM enters the fully connected DNN layer, as shown in Figure 1. The network structure was first proposed by Sainath et al. [12] in 2015 and is now widely used in the field of speech recognition.

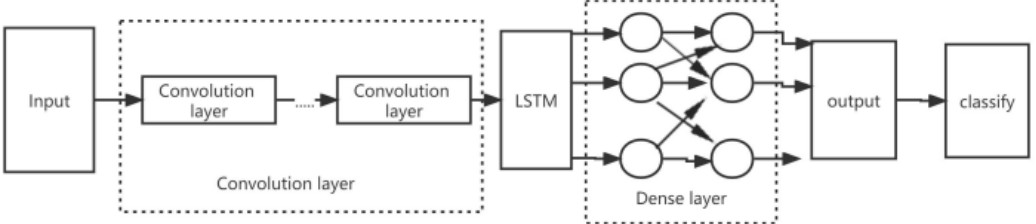

**Figure 1.** CLDNN structure.

For modeling ability, CNN (revolutionary neural network) can abstract and extract features from data at multiple time points to reduce invalid data; the LSTM (long short-term memory) structure is good at processing time-related data; DNN (deep neural network) maps the features in the input data to a more discrete space, that is, the input data become various parameters in the neural network. CLDNN model integrates the CNN layer, LSTM layer, and DNN layer into a network model in series, centralizes the advantages of each network model, and obtains better performance than a single network.

## 3. ASCLDNN Model

### 3.1. The CLDNN Network Model for Adaptive Modulation Signal Recognition Is Established

CLDNN combines the excellent performance of CNN and DNN. On the premise of ensuring accuracy, the linear layer is used to greatly reduce the amount of computation. At first, CLDNN was widely used in the field of speech recognition. For the modulation signal, this paper optimizes the CLDNN network model to make it more suitable for modulation signal recognition.

T.N. Sainath [13] mentioned that the better features of LSTM input will improve its performance. Inspired by this, after the signal enters the CNN network, this paper makes a short connection between the output of the first layer of the larger convolution layer and the later convolution layer so as to strengthen the transmission and reuse of signal features. The feature of each LSTM layer is extracted and cascaded to obtain a better effect.

### 3.2. Attention Mechanism

In recent years, researchers have introduced the selective attention mechanism of human vision [14] into deep learning. The attention mechanism is widely used in various complex deep learning tasks, such as natural language processing [15], image recognition,

and speech recognition [16]. It is one of the technologies with the greatest potential in deep learning in recent years.

The attention mechanism draws lessons from the selective attention mechanism of human vision [17]. On the one hand, its appearance reduces the computational burden of processing high-dimensional input data and reduces the data dimension by structurally selecting the subset of the input [18]. On the other hand, "eliminating the false and preserving the true" enables the task system to focus more on the current output of relevant useful information and make the allocation of resources more reasonable to improve the quality of output and improve the speed and accuracy of network recognition.

The essence of the attention mechanism is shown in Figure 2.

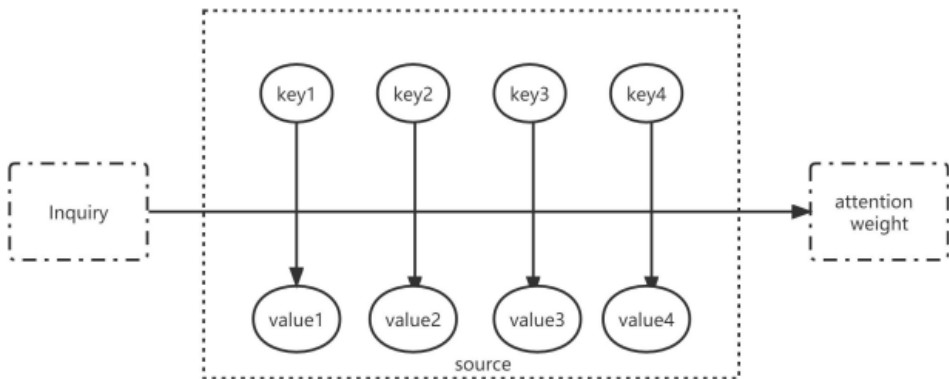

**Figure 2.** Calculation flow of attention mechanism.

Imagine that the constituent elements in the source in Figure 2 are composed of a series of <keyValue> data pairs. At this time, for an element query in the target, the weight coefficient of the value corresponding to each key is obtained by calculating the similarity or correlation between the query and each key, and then the value is weighted and summed to obtain the final value of attention [19]. Therefore, in essence, the attention mechanism is used to weight the sum of the value of the element in the source, and the query and key are used to calculate the weight coefficient of the corresponding value.

The query (q) in the attention mechanism represents the radio signal, and the key-value pair (KI VI) in the attention mechanism represents the weight of each element in the radio signal. There are three main steps in calculating attention.

Step 1, Calculate the similarity of the weights of each element in the radio signal to obtain the weights.The commonly used similarity functions are dot product, splicing, perceptron, and so on.This paper adopts the splicing method to realize the similarity [20]:

$$f(Q, K_i) = W_a[Q; K_i] \tag{1}$$

Step 2, Use the softmax function to normalize the weight obtained in the previous step:

$$\mathrm{softmax}(f(Q, K_i)) = \frac{\exp(f(Q, K_i))}{\sum_j \exp(f(Q, K_j))} \tag{2}$$

Step 3, The weight of each element in the weighted radio signal are weighted and summed to obtain attention:

$$\mathrm{Attention}(Q, K, V) = \sum a_i V_i \tag{3}$$

### 3.3. ASCLDNN Structure

After optimizing the CLDNN model, this paper adds an attention mechanism to establish the ASCLDNN network model. As shown in Figure 3, ASCLDNN is composed of the following four parts: (1) three-layer short connection convolution layer; (2) attention layer (3) one LSTM layer; (4) two-layer full connection layer; and (5) classifier.

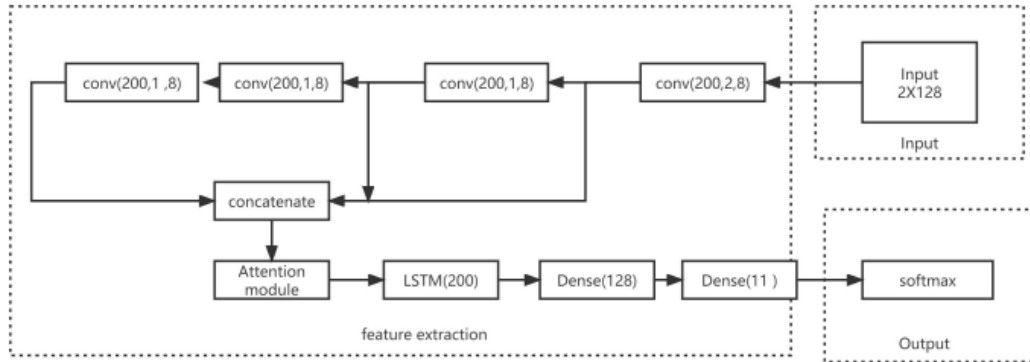

**Figure 3.** ASCLDNN network model.

The input data determined by the network input layer are the IQ bidirectional time-domain signal, and the size of each sample data is $2 \times 128$.

First part, The original data are first input to the CNN network. The first convolution layer consists of 200 layers with a size of $2 \times 8$. The feature output of this layer is short connected with the latter two convolution layers to strengthen the transmission and reuse of features. The convolution layer uses the Relu function as the activation function. The output of each convolution layer is cascaded at the end of the first part.

Second part, The convolutional neural network extracts the features of the signal and inputs them to the attention layer. The attention layer will allocate more attention resources to important features and strengthen the extraction of signal features to improve the recognition rate and training speed of modulation signal recognition.

Third part, After extracting the depth feature of the original input signal, it is input to the LSTM layer to fully extract the timing feature of the signal.

Fourth part, The signal processed by the LSTM layer is output to the full connection layer. The DNN [21] layer can map the signal features to the more discrete Euclidean feature space and obtain the deep feature information of the data. The signal processed by short connection convolution layer and attention mechanism makes the DNN network more efficient in feature extraction and improves the recognition rate and training speed of modulation signal recognition.

Fifth part, The final result is input into the softmax layer to reduce the dimension of the signal. Set the node size of the full connection layer in front of the softmax layer to 11 and output it in the form of an 11-dimensional probability vector. The index of the maximum probability value is used as the classification result.

At the same time, the loss function of the network is defined as the loss function in the cross-layer. In addition, to prevent the phenomenon of overfitting in training, dropout regularization technology is adopted to make the neurons of each convolution layer have a 1/2 probability of maintaining the current state and no longer participate in the calculation of forwarding transfer and reverse transfer.

*3.4. Dataset*

This paper adopts the international general dataset RML201610A [22]. The dataset contains 8 types of digital modulation signals (BPSK, 8PSK, CPFSK, GFSK, pam4, qam16, QAM64, and QPSK) and 3 types of analog modulation signals (AM-DSB, am-SSB, WBFM), a total of 11 different types of modulation signals. The signal-to-noise ratio of 11 signals is distributed in the range of −20–18 db, with an interval of 2 dB. Under different signal-to-noise ratios, each signal is composed of two IQ channels, with 128 sampling points. In addition, to be close to the real environment, this dataset adds influencing factors, such as center frequency shift, fading exit, and additive Gaussian white noise to various signals.

RML201610A is used in the experiment to compare and analyze the performance of the ASCLDNN network. There are 221,000 samples in the 10 A data set, of which 70% are used as the training set and 30% as the test set. (It was tested that 70% of the input signal is

sufficiently trained when chosen as the training set. If the proportion of the training set is increased further, the results obtained are not improved). The network models in this paper adopt the theano backend and keras framework. The hardware configuration is Intel i7-8750cpu, GPU NVIDIA GeForce rtx2060.

## 4. Experimental Results and Model Parameter Optimization

### 4.1. Influence of Attention Mechanism on Network Classification Ability

To explore the impact of the added attention mechanism on the network classification ability, we will compare the models with and without the added attention mechanism by taking the training time, training accuracy, loss rate, and other parameters of the network model on the test set as the evaluation criteria. Specific data are shown in Table 1:

**Table 1.** Effects of adding attention mechanism on the model.

| Model | Training Time (s) | Loss % | Training Accuracy % |
|---|---|---|---|
| SCLDNN | 0:11:22 | 0.405 | 60.7 |
| ASCLDNN | 0:3:34 | 0.329 | 64.2 |

It can be seen from the table that after adding the attention mechanism, the training time of the model is significantly shortened, the loss rate of training is reduced, and the accuracy is improved to a certain extent. The attention mechanism provides a set of effective solutions to the problem of model information overload. It can extract more important information for the current task from a great deal of information, pay high attention, and ignore other irrelevant information to improve the efficiency and accuracy of the model for task solving. Next, we will continue to optimize the ASCLDNN model by adjusting the model parameters, including the number and size of convolution cores and the model structure, such as the number of convolution layers and the number of LSTM layers.

### 4.2. Influence of Convolution Check on Recognition Performance

We test the convolution kernel sizes of $1 \times 6$, $1 \times 7$ and $1 \times 8$, respectively.

Fixed convolution kernel sizes are $1 \times 6$, $1 \times 7$, and $1 \times 8$. Change the number of convolution kernels between 50 and 1000, analyze, observe and compare the impact of different convolution kernels on the recognition effect, and find a kernel number with the best recognition performance under the current convolution kernel size. The recognition effect under different convolution kernels is obtained by experiment, as shown in Figures 4–6.

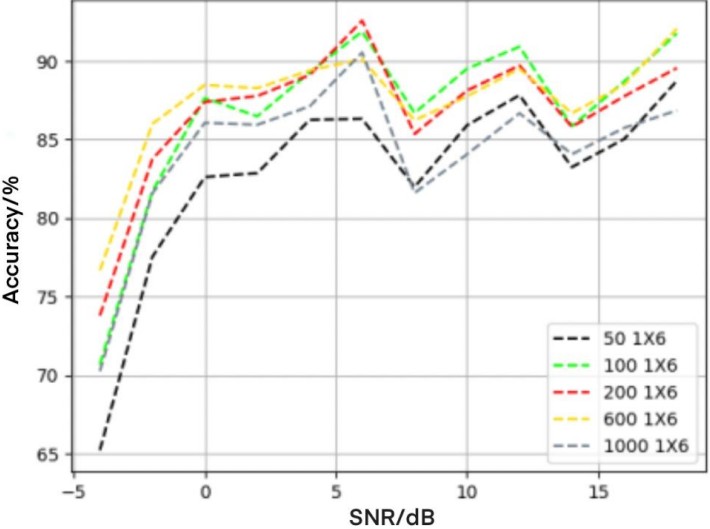

**Figure 4.** Effect of different convolution kernels on recognition rate (size $1 \times 6$).

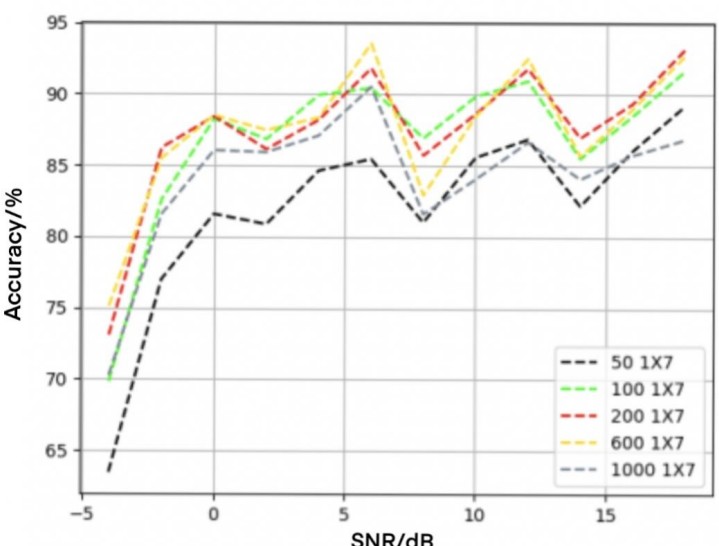

**Figure 5.** Effect of different convolution kernels on recognition rate (size $1 \times 7$).

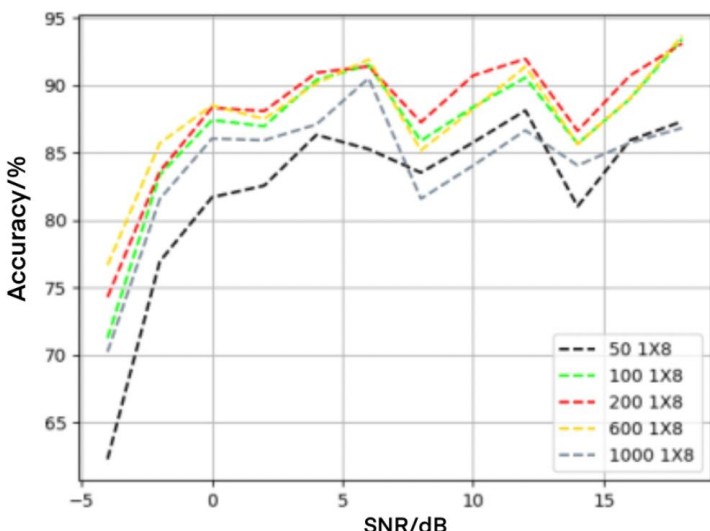

**Figure 6.** Effect of different convolution kernels on recognition rate (size $1 \times 8$).

It is found that under the condition of different convolution kernel sizes, the change of the number of convolution kernels has a similar impact on the final accuracy. In the process of increasing the number of convolution kernels from 50 to 200, the recognition rate has an upward trend. When more than 200 convolution kernels are added, the recognition rate will slow down, the training time will increase, and the training efficiency will be reduced. Therefore, it can be determined that the peak of the recognition effect occurs when the number of convolution cores is 200.

By comparing the recognition rates of 200 convolution kernels in $1 \times 6$, $1 \times 7$, and $1 \times 8$, we finally choose 200 $1 \times 8$ convolution kernels as the optimal case.

### 4.3. Influence of Convolution Layers on Recognition Performance

According to the experimental results in 3.2, the number of convolution cores of each convolution layer in the fixed CLDNN network is 200 and the size is $1 \times 8$. Other structures of the model remain unchanged. Starting from the 1-layer convolution layer network, increase the number of convolution layers step by step, and keep the number and size of the convolution cores increased to 200 and $1 \times 8$. Observe the influence of the different convolution layers on the model recognition performance.

The final result is shown in Figure 7. We test the case of 1–5 convolution layers. It can be seen that when the number of convolution layers is 4, the model has the highest and most stable recognition rate. When the number of layers is less than 4, the features extracted by the model are insufficient. When the number of layers is greater than 4, the model will extract redundant features and improve the complexity of the model.

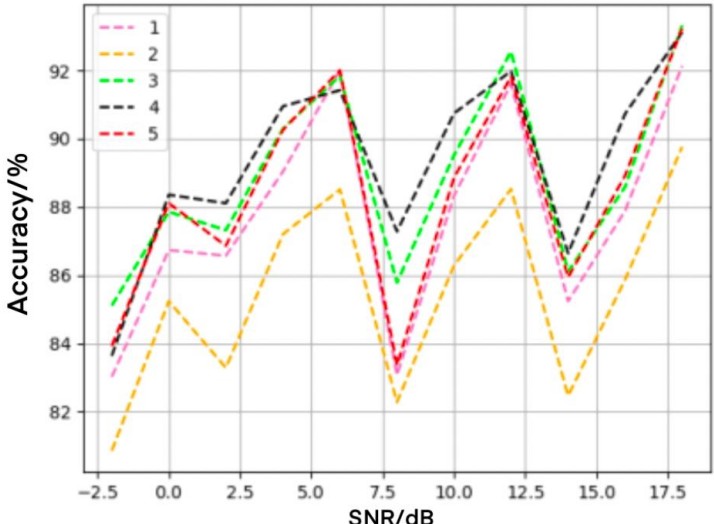

**Figure 7.** Effect of different convolution layers on recognition rate.

### 4.4. Influence of LSTM Layers on Recognition Performance

LSTM, namely long-term and short-term memory network, is an improved cyclic neural network. It is very effective in solving the common problems of gradient explosion and gradient disappearance in the neural network. The LSTM layer can extract the timing features of signals. Too few LSTM layers may lead to incomplete timing feature extraction, and too many layers will increase the complexity of the network and reduce the recognition performance. Fix the network structure determined by the experiment before, observe the recognition performance of the network by changing the number of LSTM layers, and set the LSTM output size of the first layer to 250. When the number of layers is increased later, the LSTM output size is set to 128.

Finally, the results are shown in Figure 8 and Table 2. With the increase in the number of LSTM layers, the number of parameters that need to be trained in the network also increases greatly. At the same time, the training of the model increases correspondingly with the classification time on the test set, but the classification performance is declining. The results in Table 2 show that the average recognition rate and maximum recognition rate of the network model with one layer of LSTM are higher than those of multiple layers in a high signal-to-noise ratio. It can be seen that the time-series characteristics of the signal extracted by one layer of LSTM can meet the requirements of classification.

**Table 2.** Influence of LSTM layers on model recognition rate.

| LSTM Layers | 1 | 2 | 3 |
|---|---|---|---|
| Average recognition rate of high signal-to-noise ratio | 89.93 | 88.91 | 88.66 |
| Highest recognition rate | 93.12 | 92.92 | 92.83 |

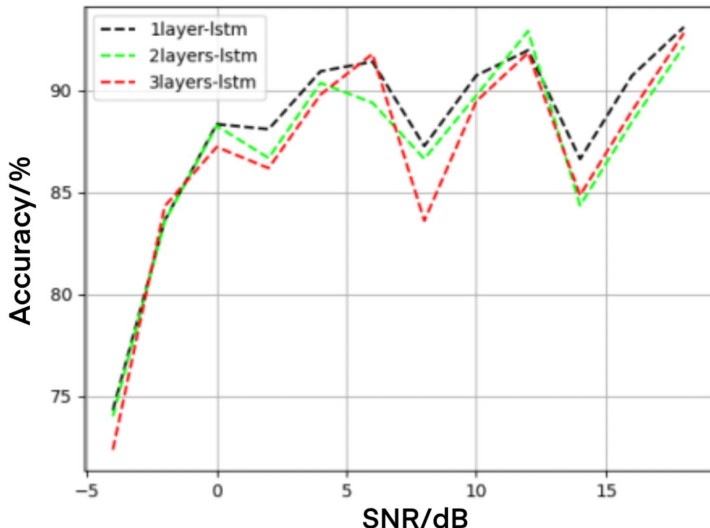

**Figure 8.** Effect of different LSTM layers on recognition rate.

*4.5. Performance Analysis of Different Networks*

Experimental analysis and comparison is conducted of the CLDNN network in this paper, an optimized CLDNN network in ref. [3], DNN network in ref. [1], Google net network, and Alex net network. The effect of various network models on classification tasks is shown in Figure 9.

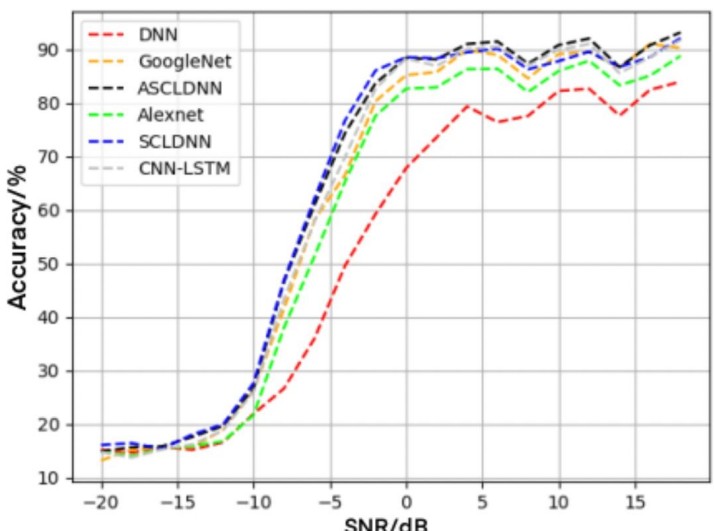

**Figure 9.** Comparison of recognition performance of six models (RML2016.10a).

As shown in Figure 9, even compared with the current newer model, the recognition rate of the model proposed in this paper is significantly improved in the range of −6 dB to 0 dB, and the recognition rate is relatively high in the case of the high signal-to-noise ratio. The following table analyzes and compares the specific performance data of ASCLDNN with other models.

It can be seen from Table 3 that the recognition speed and accuracy of short connection of the convolution layer based on CLDNN are faster and higher than that of the common CLDNN network and SCLDNN network with the only short connection of the convolution layer.

**Table 3.** Training classification effect of various models.

| Model | Training Time (s) | Number of Training Rounds | Loss % | Training Accuracy % |
|---|---|---|---|---|
| ASCLDNN | 0:3:34 | 17 | 32.9 | 64.2 |
| SCLDNN | 0:11:22 | 26 | 40.5 | 61.7 |
| GOOGLENET | 0:16:09 | 25 | 34.3 | 60.8 |
| ALEXNET | 0:05:39 | 5 | 26.1 | 58.9 |
| DNN | 0:7:41 | 40 | 12.9 | 54.6 |
| CNN-LSTM | 0:11:09 | 34 | 33.8 | 60.2 |

Figures 10 and 11 are the confusion matrix diagrams of CLDNN and ASCLDNN models when the signal-to-noise ratio is −2 dB, which can intuitively reflect the recognition effect of the model on 11 signal modulation modes. According to the previous experimental results, when the signal-to-noise ratio is greater than −2 dB, the recognition rate of various models for various signals fluctuates, but it is basically stable in a certain range, so we choose the signal-to-noise ratio of −2 dB for analysis. It can be seen from the figure that the CLDNN model mistakenly identifies part of WBFM as AM-DSB and confuses the part of 8PSK and QPSK, resulting in a low recognition rate of 8PSK, WBFM, and QPSK signals. The ASCLDNN model does not recognize part of WBFM as AM-DSB and reduces the confusion of the 8PSK and QPSK signal modulation methods. It can be predicted that in the future, the focus of improving the network recognition rate is to continue to reduce the confusion of 8PSK and QPSK modulation methods.

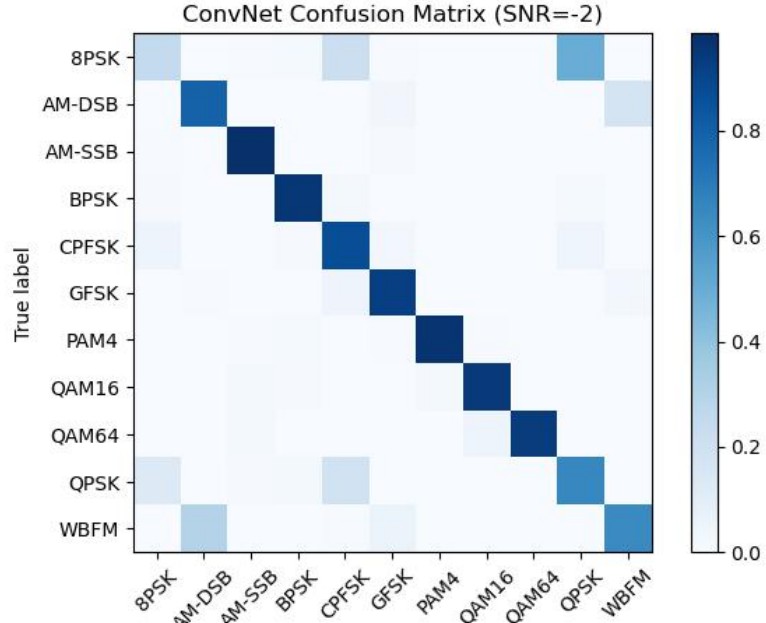

**Figure 10.** Confusion matrix on SNR = −2 dB (SCLDNN).

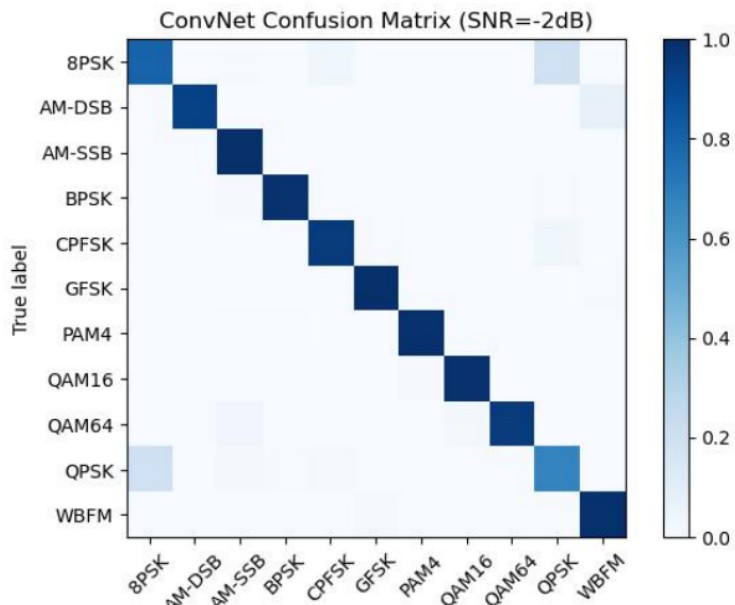

**Figure 11.** Confusion matrix on SNR = −2 dB (ASCLDNN).

## 5. Conclusions

In this paper, based on the CLDNN model, the CLDNN end-to-end network model applicable to modulated signal awareness ASCLDNN is proposed, and various types of mathematical modulated signals and analog modulated signals are identified and studied. The experimental results show that short connection optimization and the addition of an attention mechanism based on the CLDNN network are very effective. The ASCLDNN network has the advantage of less training time and training rounds but better accuracy when compared with other, newer network models.

The ASCLDNN network can recognize 11 types of signal modulation with high recognition accuracy, even when the signal-to-noise ratio is low and no confusion is generated for specific signals. At the same time, the ASCLDNN model proposed in this paper can automatically recognize all kinds of modulated signals without human participation, and it can be free from interference when the noise is high, which has the advantages of fast training and high recognition accuracy compared with existing methods.

**Author Contributions:** Conceptualization, B.Z. and X.Z.; formal analysis, B.Z. and F.W.; data curation, B.Z. All authors have read and agreed to the published version of the manuscript.

**Funding:** This research received no external funding.

**Conflicts of Interest:** The authors declare no conflict of interest.

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
