# Peer review of "Research on Modulation Signal Recognition Based on CLDNN Network"

_electronics, doi:10.3390/electronics11091379_

Round 1
Reviewer 1 Report
Comments and Suggestions for Authors:
This paper addressed the study of modulation signal recognition based on CLDNN network. It is working on browsing the effects of adding attention mechanism on the model. The manuscript focused on the preferability of using attention-mechanism short-link convolution long-short-term -memory deep neural networks (ASCLDNN) on the short-link convolution long-short-term -memory deep neural networks (SCLDNN). The obtained results browsed the accuracy increasing when using (ASCLDNN) over those of (SCLDNN).
The paper has significant issues in the research experimental simulation. I hope that my comments will help to improve the quality of the article:
Recheck the English language level of the paper, and avoid using long sentences.
Abstract:
- This is a very long sentence “Aiming at the shortcomings of existing recognition methods, such as high manual participation, few recognition types, and low recognition rate under low signal-to-noise ratio, an attention-mechanism short-link convolution long-short-term -memory deep neural networks (ASCLDNN) recognition model is proposed”, is a very long one, as shown in lines (3 to 6). Please rewrite the Abstract in the correct form.
- Don't talk about practical steps, but you should talk about the depended algorithms and models. , as shown in lines (6 & 7).
- The existence of significant numerical values for the design, as “recognize 11 signal modulation modes,”, is a good step toward brows strength of the work. But, it is better to add significant numerical values for the outcomes also.
1. Introduction:
- In this section, you should explain clearly the main aim of the manuscript, the important questions that must be answered in the following sections, and the main contributions).
- You used three references for three lines (as shown in line 21), which is too much. So, try to use fewer references that are more related to your subject.
- Let your contribution be soundest than be expressed in a simple sentence to support your work (as shown in lines 38 to 40).
- Please move the following text:
“Experiments show that compared with the existing methods, the training time and accuracy of the network model”, as shown in lines (40 to 42) to the conclusion section.
- CLDNN Model:
- In line 44, the C in (CLDNN) means “Convolutional” not “Continuous”, which leads to the full name of (CLDNN) to be “Convolutional Long- Short Term Deep Neural Network”.
- Figure 1 is unclear, so redraw it, not copy-paste it from [12].
- Where is the reference to the paragraph shown in lines (50 to 56)?
- ASCLDNN Model:
- Where is the paragraph's reference shown in lines (59 to 62)?
- Avoid using more than one reference for one speech, as shown in line 72.
- Where is the reference to the paragraph shown in lines (75 to 81)? Otherwise, how do you conclude it?
- Figure 2. is unclear, so redraw and write the Reference number.
- Where is the reference to the paragraph written below the Figure 2?
- Please write the References of all equations.
- Rewrite equation 2 according to Mathematical principles, not programming style.
- What are the starting and ending values of variable i in equation 3?
- Figure 3 is unclear, so redraw it, not copy-paste it from the reference, and write the reference number.
- In lines (133 to 134), you talked about the Training and Testing ratios. Did you check other ratios for Training and Testing, and where are their results if any?
- Experimental Results and Model Parameter Optimization
- Explain the meaning of Training Accuracy, which appeared in all Tables?
- Are the obtained values of Training Accuracy in Table 1 acceptable?
- Did you combine all four parameters (Training time, Number of training rounds, Loss, and Training accuracy) in Table 3 to make the correct decisions related to the best models?
- Why did you select exact these values of rounds in Table 3?
- Are the obtained values of Training Accuracy in Table 3 acceptable?
- You should support your outcomes by comparing them with those of the closest previous works in this field.
- Conclusion:
Write the conclusion in the past tense. The conclusion is insufficient to cover the proposed models and outcomes, so it needs more explanation. Adding to that, please add some important numerical results compared to previous works.
- Reference:
The references are good and up to date.
Author Response
Dear reviewers,
Thank you very much for your kindly comments on our manuscript. There is no doubt that these comments are valuable and very helpful for revising and improving our manuscript. In what follows, we would like to answer the questions you mentioned and give detailed account of the changes made to the original manuscript.
Abstract:
A
Thank you for your significant reminding. According to your suggestion, we corrected the above grammatical errors and made an effort to correct the spelling and grammar errors and polish the whole manuscript. We would like to confirm that the suitably revised manuscript is understandable to readers.
I have rewritten this long sentence using the correct format. The overall language of the abstract was also touched up.
B
Thanks for your valuable counsel.
I have succinctly described the algorithms and models relied upon in the abstract.
C
Thank you for your instructive comments. I have added a discussion on the confusion generated by various signals to enrich the final result.
- Introduction:
A
Thanks for your valuable counsel.
After revising the last paragraph, I clearly expressed the main purpose of the manuscript, the important questions that must be answered in the following sections, and the main contributions.
B
Thanks to your comments, I used fewer references related to my topic, while adding more citations that expand the direction.
C
Thanks for your valuable counsel.We have described our work in detail, adding credibility.
D
Thanks for your valuable counsel.The summary language has been moved to the conclusion section.
- CLDNN Model:
A
Thank you for pointing out this detail error, we have corrected it. We are deeply sorry for our oversight.
B
Thanks for your valuable counsel.Figure 1 is our own production. It is only because the manuscript has been reworked many times and constantly copying it caused it to become blurry. We recreated the picture.
C
Thanks for your valuable counsel.We have modified the reference number position.
- ASCLDNN Model:
A
Thanks for your valuable counsel.We have modified the reference number position.
B
Thanks for your valuable counsel.I have revised the relevant reference numbers based on your comments.
C
Thanks for your valuable counsel.I have correctly cited the references for clarification.
D
Thanks for your valuable counsel.Figure 2 is our own production. It is only because the manuscript has been reworked many times and constantly copying it caused it to become blurry. We recreated the picture.
E
Thanks for your valuable counsel.We have modified the paragraph structure, which is illustrated below Figure 2.
F
Thanks for your valuable counsel.Added relevant references.
G
Thanks for your valuable counsel.Rewrote Equation 2.
H
Thanks for your valuable counsel. The variable i corresponds to the weight of each element in the radiation source signal.
I
Thanks for your valuable counsel.Figure 3 is our own production. It is only because the manuscript has been reworked many times and constantly copying it caused it to become blurry. We recreated the picture.
J
Yes. We have tested other training ratios. The best results were obtained when using 70% of the data as the training set. It has been tested that 70% of the input signal has been sufficiently trained when chosen as the training set. If the proportion of the training set is increased further, the results obtained will not be improved.
- Experimental Results and Model Parameter Optimization
A
It refers to the global accuracy of the model for all input signals, including 11 different signals from -18dB to 20dB.
B
This is acceptable. Because when the signal-to-noise ratio is very low, the input signal is almost all noise, and the recognition accuracy of various models will be very low. Only after the signal-to-noise ratio is gradually increased to 0bB, the recognition rate of the model for the input signal will be increased to a relatively high degree. The accuracy rates in Table I refer to the global accuracy rates, which are the average recognition rates of various signals on -18dB to 20dB. The recognition rate for signals below 0db is often very low, well below 50%. So in the final global recognition rate, to exceed 60% is already a very good performance.
C
Yes. We took into account all factors to select the best model.
D
To be honest, because we use the python language, we can see the exact number of training rounds in the run results screen.
E
This is acceptable. Because when the signal-to-noise ratio is very low, the input signal is almost all noise, and the recognition accuracy of various models will be very low. Only after the signal-to-noise ratio is gradually increased to 0bB, the recognition rate of the model for the input signal will be increased to a relatively high degree. The accuracy rates in Table I refer to the global accuracy rates, which are the average recognition rates of various signals on -18dB to 20dB. The recognition rate for signals below 0db is often very low, well below 50%. So in the final global recognition rate, to exceed 60% is already a very good performance.
F
Your suggestions are very useful. We have compared the newer models in Figure 9. Comparatively, our model works better.
- Conclusion:
Thanks for your valuable counsel.We have added more explanations and also added some important numerical results compared to previous work.
- Reference:
We have also refined the references.
Thank you again for your positive and constructive comments and suggestions on our manuscript.
We hope you will find our revised manuscript acceptable for publication.

Reviewer 2 Report
In this article the authors are proposing an end-to-end network model which studies the recognition of various digital and analog modulation signals.
Overall, the article is presenting an actual interesting topic. However, the authors should provide several reasons why their model introduces a confusion between QPSK and 8PSK (line 216).
Line 230. Could the authors describe some practical applications “The experimental results also prove that the model has practical 230 application value.”?
Author Response
Dear reviewers,
Thank you very much for your kindly comments on our manuscript. There is no doubt that these comments are valuable and very helpful for revising and improving our manuscript. In what follows, we would like to answer the questions you mentioned and give detailed account of the changes made to the original manuscript.
Overall, the article is presenting an actual interesting topic. However, the authors should provide several reasons why their model introduces a confusion between QPSK and 8PSK (line 216).
Thank you for pointing this out.From the confusion matrix diagram, we can see that the conventional CLDNN model incorrectly identifies some WBFM as AM-DSB and confuses some 8PSK, QPSK, which leads to a low recognition rate of the three signals 8PSK, WBFM and QPSK. This is because the 8PSK, QPSK signals are similar and it is difficult for the neural network model to learn the difference between these two types of signals.
Our proposed ASCLDNN model optimizes the short connection layer and adds an attention mechanism to assign higher weights to important signals during the learning process, which makes it learn 8PSK and QPSK signals better and does not incorrectly identify some WBFM as AM-DSB, and reduces the confusion between 8PSK and QPSK signal modulations.
Line 230. Could the authors describe some practical applications “The experimental results also prove that the model has practical 230 application value.”?
Thanks for your valuable counsel.The ASCLDNN model proposed in this paper can automatically identify all kinds of modulated signals without human participation, and it can be free from interference when the noise is high, which has the advantages of fast training and high recognition accuracy compared with existing methods. Therefore, we believe that this model may have some practical value, but it is not currently involved in practical projects. Thanks for your reminder, we deleted this sentence“The experimental results also prove that the model has practical 230 application value.”.
Thank you again for your positive and constructive comments and suggestions on our manuscript.
We hope you will find our revised manuscript acceptable for publication.
